

# ER3D: a structural and geophysical 3D model of central Emilia-Romagna (Northern Italy) for numerical simulation of earthquake ground motion

Peter Klin[1], Giovanna Laurenzano[1], M. Adelaide Romano[1], Enrico Priolo[1], Luca Martelli[2]

[1]Centro Ricerche Sismologiche (CRS), Istituto Nazionale di Oceanografia e di Geofisica Sperimentale (OGS), Sgonico (TS), 34010, Italy

[2]Servizio Geologico Sismico e dei Suoli, Regione Emilia-Romagna, Bologna, 40127, Italy

*Correspondence to*: Peter Klin (pklin@inogs.it)

**Abstract.** During the 2012 seismic sequence of Emilia region (Northern Italy), the earthquake ground motion in the epicentral area featured longer duration and higher velocity than those estimated by empirical-based prediction equations typically adopted in Italy. In order to explain these anomalies, we (1) build up a structural and geophysical 3D digital model of the crustal sector involved in the sequence, (2) reproduce the earthquake ground motion at some seismological stations

through physics-based numerical simulations and (3) compare the observed recordings with the simulated ones. In this way we investigate how the earthquake ground motion in the epicentral area is influenced by local stratigraphy and geological structure buried under the Po Plain alluvium. Our study area covers approximately 5000 km² and extends from the Po river right bank to the Northern Apennines morphological margin in N-S direction, and between the two chief towns of Reggio Emilia and Ferrara in W-E direction, involving a crustal volume with 20 km of thickness. We set up the 3D model by using

already published geological and geophysical data, with a detail corresponding to a map at scale 1:250,000. The model depicts the stratigraphic and tectonic relationships of the main geological formations, the known faults and the spatial pattern of the seismic properties. Being a digital vector structure, the 3D model can be easily modified or refined locally for future improvements or applications. We exploited high performance computing to perform numerical simulations of the seismic wave propagation in the frequency range up to 2 Hz. In order to get rid of the finite source effects and validate the model

response, we choose to reproduce the ground motion related to two moderate-size aftershocks of the 2012 Emilia sequence that were recorded by a large number of stations. The obtained solutions compare very well to the recordings available at about 30 stations, in terms of peak ground velocity and signal duration. Snapshots of the simulated wavefield allow us to explain the exceptional length of the observed ground motion as due to surface waves overtones that are excited in the alluvial basin by the buried ridge of the Mirandola anticline. Physics-based simulations using realistic 3D geo-models show

eventually to be effective for assessing the local seismic response and the seismic hazard in geologically complex areas.

## 1 Introduction

Computer aided three-dimensional (3D) geological modeling (e.g. Mallet, 2002) is becoming an increasingly important tool in geo-science studies for both the management of natural resources and the prevention of natural disasters. 3D geological modeling allows the combination of multi-disciplinary data in the shaping and visualization of the current knowledge of the

geological structures and allows the integration with new data or interpretations, as they become available (Calcagno, 2015). Moreover, 3D geological models represent the basis for the execution of physics-based numerical simulations, provided that a reliable scientific procedure is defined to convert the different types and levels of the available complex geological information into that needed by the proposed numerical simulation at the predefined scale level (e.g. Fischer et al., 2015).




The present study concerns the development of a structural and geophysical 3D model to be used in physics-based numerical simulations of seismic wave propagation aimed at explaining the observed ground motion in past earthquakes and at increasing the reliability of ground motion predictions for possible future events (e.g. Moczo et al., 2014; Taborda and Roten, 2015; Cruz-Atienza et al., 2016). Our study focuses on the Emilia region (Northern Italy), where in 2012 a relevant

seismic sequence featuring the two main shocks $M_W$ 6.1 on 20/5/2012 at 02:03:53 UTC and $M_W$ 5.9 on 29/5/2012 at 07:00:03 UTC (Rovida et al. 2016), occurred (Fig. 1).

Seismic-hazard studies are usually based on the empirical-statistical method, which makes use of ground motion prediction equations (GMPE) (e.g.: Barani et al., 2017a and 2017b), with possible corrections deduced from local geological conditions (Grelle et al., 2016). However, occasionally the observed ground motion characteristics deviate considerably from the

empirical-statistical predictions. Those deviations are usually due to physical phenomena that in principle can be taken into account by using the numerical-deterministic method.

An emblematic case of such deviations occurred during the 2012 Emilia seismic crisis, when unexpectedly long duration and large peak ground velocity (PGV) characterized the earthquake ground motion at some sites in the epicentral area (Castro et al., 2013; Luzi et al., 2013; Barnaba et al., 2014; De Nardis et al., 2014). Those deviations have been attributed to the

complexity of the geological structure beneath the Po Plain, which features a very large and deep alluvial basin bounded by two largely buried thrust-and-fold systems, the Northern Apennine chain at South and the Southern Alpine ridge at North, respectively (Boccaletti et al., 1985). In order to explain quantitatively the observed ground motion characteristics, we have developed a 3D model that describes the morphology of the buried geological structure and assigns visco-elastic properties (mass density, elastic modula and elastic quality factors), to each formation, so that it can be used for physics-based

numerical modeling of the seismic wave propagation in the studied volume.

Our model is not the first 3D model that was developed for the Po Plain area. At least three research groups have carried out 3D numerical computations of the earthquake ground motion in the Po Plain so far. A first study was performed by Vuan et al. (2011), who simulated long-period (T > 5 s) surface waves generated in the basin by strong ($M_W$ > 6) earthquakes. They used a 3D model featuring realistic, irregular basin edges and a simplified depth-dependent velocity profile for the

sedimentary filling of the basin. A more complex 3D geological model was set-up by Molinari et al. (2015) for simulating the earthquake ground motion in the long-period band (T > 3 s). The simulated waveforms were compared only qualitatively with the recorded waveforms at some far-source stations in order to demonstrate the effectiveness of the 3D geological model. A third model is that one developed by Paolucci et al. (2015), who simulated the near-source strong ground motion for the $M_W$ 6.1 20 May 2012 earthquake in the frequency range 0.1-1.5 Hz. The overall satisfactory agreement of their

simulated waveforms with the empirical records was however attributed principally to the assumed extended source model (i.e. slip distribution and rupture propagation) rather than to their model structure, which contains only two main geologic interfaces. We have to mention also Turrini et al. (2014), who defined the whole structure of the Po basin from its deep roots, at the Moho level, through an exhaustive analysis of all the existing structural-geological and geophysical studies. They summarize the current knowledge of the Po basin structural-geology into a digital, editable model that can be used to

improve the geodynamic interpretation of the area. However, their model does not contain any geophysical parameterization, neither it reaches the level of detail that is required in our study.

In the present work we focus on a more detailed 3D geological model of a limited area of the Po Plain, bounded by the Po river right bank at North, by the Northern Apennines morphological margin at South, and located between the two chief towns of Reggio Emilia at West, and Ferrara at East (Fig. 1). The area includes the epicenters of the 2012 seismic sequence

as well as some other potential seismogenic structures (DISS Working Group, 2015). In order to set up the 3D geological model we considered only data available in scientific literature. The physical properties assigned to the geological units were deduced from literature as well. We relied on the commercial software GeoModeller released by Intrepid Geophysics for

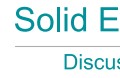
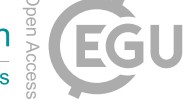

merging and interpolating the geological data in a 3D digital model, which constitutes the input for numerical simulations of the earthquake ground motion and represents a basis for further improvements when new data will be available. In this first version, denoted ER3D, the model is based on the elaboration of a Digital Terrain Model, a seismotectonic map, three deep geological sections crossing the study area as well as the isobaths of two interfaces between some relevant geological

formations. The detail level included in this model is consistent with numerical computations of the ground motion in the frequency range up to 2 Hz, therefore comparable with the frequency range of the computations performed by Paolucci et al. (2015).

We performed the computations of the ground motion by applying the high-performance computing (HPC) code FPSM3D (Klin et al., 2010), dedicated to the numerical modeling of the propagation of viscoelastic waves in heterogeneous media.

The code FPSM3D is based on the Fourier pseudo-spectral method for the solution of hyperbolic equations; its accuracy performance compares well with other computer codes used in the scientific community for the 3D simulation of the earthquake ground motion in alluvial basins, as emerged during recent verification exercises (Maufroy et al., 2015; Chaljub et al., 2015). The validation of the constructed 3D geological model consisted in a comparison in terms of PGV and duration, between the numerical predictions and the empirical recordings of two 2012 events at the several stations that were deployed

in the area during the seismic sequence (Fig. 1). In order to put in evidence the peculiarities of the ground motion that are due only to propagation effects, we considered two weak events ($M_W$ 4.0 and 4.1), that can be modeled using a point source, and not the mainshocks, which would require a finite source model. The computations were run using the HPC resources of the CINECA consortium in Bologna.

**2 The structural and geophysical 3D model of the central Emilia**

The fundamental step for physics–based numerical prediction of the earthquake ground motion consists in the set-up of a 3D model of the geological structure. In order to set up of a reliable geological model we need a sound geological interpretation of well-constrained geophysical data. Thanks to oil exploration and research widely undertaken since 1960, a comprehensive synthesis of the structural features of the Po Plain subsurface was possible in the past decades (e.g. Pieri and Groppi, 1981; Fantoni and Franciosi, 2010; Boccaletti et al., 20111; Martelli et al., 2017). In the following we give an overview of the

known geological features of the study area and describe how we synthetized these data in a digital 3D structural model. Finally, we discuss how we assigned the physical properties to each geological formation for characterizing the 3D model also from a geophysical point of view.

**2.1 Geological and seismotectonic setting of the study area**

The study area is in the Emilia-Romagna region (Northern Italy), and specifically it occupies the sector of the Po Plain

between Reggio Emilia (West) and Ferrara (East), as shown in Fig. 1. The Po Plain is a foredeep-foreland zone interposed between two chains with opposite vergence: Northern Apennines to the South and Southern Alps to the North. Terrigenous sediments originating from the erosion of the two growing chains accumulated in the basin (Dondi et al., 1982), first those of alpine origin (Miocene-Quaternary) then those of Apennine provenance (Pliocene-Quaternary). From the Middle Pleistocene the sedimentation is mainly continental and results from the depositional activity of the Po River and its tributaries. The

substrate of the terrigenous sediments is made up by a carbonatic succession of mainly Mesozoic age, whose top consists of marly sediments of Paleogene age. This carbonate succession is separated from the metamorphic basement by a thick evaporite succession of Triassic age (Fig. 2). From the tectonic point of view, the area is affected by numerous compressive structures, with northern vergence (Fig. 1). The southern zone, coinciding with the Apennines hills between Albinea, Sassuolo, Vignola and Casalecchio di Reno municipalities, is characterized by the reverse faults of the Pedeapenninic Thrust



(Boccaletti et al., 1985), which is responsible for the morphological transition between the Northern Apennines and the Po Plain. Subsoil investigations for oil exploration (Pieri and Groppi, 1981) showed that the Apennine outer front does not coincide with the Apennine - Po Plain morphological margin and that in the Po Plain subsoil many blind faults and folds are present. Actually, the Apennine outer front is located in the subsoil around the present course of the Po River, coinciding

with the reverse faults of the Ferrara Folds overthrusting the Lombardy-Veneto monocline (Fig. 1). The main detachment and overlap levels of thrusts are the Triassic evaporites, embedded between the underlying metamorphic basement and the overlying succession made of Late Triassic-Oligocene carbonates, Oligo-Miocene marls, and more recent terrigenous sediments (geological sections in Fig. 2). The southernmost buried structures, characterizing the subsoil of the plain between Reggio Emilia, Modena and Bologna, are the eastern termination of the Emilia Folds and the western termination of

Romagna Folds. The northernmost structures are in the subsoil between Novellara, Mirandola and Finale Emilia, where they constitute the western arc of the Ferrara Folds (Fig. 1), giving rise to a very pronounced ridge, whose top is very close to the surface between Novi di Modena and Mirandola. Large part of the interest area, in particular the central zone between Modena, Carpi and Cento, comprised between the Pedeapenninic Thrust and the Ferrara Folds, corresponds to a very deep syncline: the thickness of the Plio-Quaternary sediments between Modena and Crevalcore exceeds 8500 m (Pieri and

Groppi, 1981).

The relationships between tectonic structures, sedimentary bodies and the surface morphology indicate that Pedeapenninic Thrust and Ferrara Folds were active also in recent times, as demonstrated by the Quaternary deposits which are deformed and uplifted. Conversely, the Emilia and Romagna Folds were active mainly in the Pliocene, being the Quaternary deposits not deformed by these structures but included in the syncline between the Pedeapenninic Thrust and the Ferrara Folds (Pieri

and Groppi, 1981; Burrato et al., 2003; Boccaletti et al., 2004 and 2011; Martelli et al., 2017).

The entire area is seismically active, and the distribution of historical and instrumental earthquakes seems to confirm the major actual activity of the Pedeapenninic Thrust and the Ferrara Folds. In fact, the main historical earthquakes of the area have been located along the Apennines-Po Plain margin (Rovida et al., 2016), while Ferrara Folds are responsible for earthquakes of 20 and 29 May 2012, respectively $M_W$ 6.1 and $M_W$ 5.9 (Fig. 1). For these reasons, these structures are

included in the database of the seismogenic structures capable of generating strong earthquakes (DISS Working Group, 2015). The instrumental data indicate that in this area the most part of earthquakes has a compressional source mechanism (Pondrelli et al., 2006), and that the hypocentral depth (http://cnt.rm.ingv.it, last accessed May 2018) in the northern zone (Ferrara Folds) is concentrated in the first 15 km while in the Apennines-Po Plain margin greater depths (15-35 km) are common.

**2.2 Integration of geological data in the 3D digital model**

Geological 3D modeling consists in the representation of a solid Earth sector by using surface and subsurface data in a computer-aided process (Mallet, 2002), which allows to shape and to visualize the current knowledge and/or to update it with new data. Numerous methodologies were implemented in several packages dedicated to the geological 3D modeling. The package we adopt for the present work is GeoModeller (Calcagno et al., 2008; Guillen et al., 2004), a commercial

software originally developed by the French Bureau de Recherches Géologiques et Minières (BRGM) and more recently by the Intrepid Geophysics (http://www.geomodeller.com, last accessed May 2018). GeoModeller is a software tool for building complex, steady state and implicit 3D geological models, directly from geological observations. The interpolation method is based on the potential field theory (Lajaunie et al., 1997; Chiles et al., 2006), so that geological interfaces (i.e. the upper or lower surfaces of the geological units) are modelled as iso-surfaces of a scalar potential field defined in the 3D space.

Structural data are treated as the gradient of the field. The interpolation of the field uses cokriging to take both contacts and structural data into account and generates surfaces that honour all the data together (McInerney et al., 2005). The adopted





approach also employs rule-based modeling to control the relationships in the stratigraphic pile (either 'onlapping' or 'erosional'), and to control fault chronology within the fault network (Chiles et al., 2004). All the details about the application of this method for building geological models can be found in Calcagno et al. (2008).

To build the 3D geological model of the central Emilia we considered a crustal volume with 70 km x 70 km area and depth of 20 km, in order to include most of the 2012 seismic sequence hypocenters, associated to the deepest segments of the active thrusts. We defined the stratigraphic pile according to that one reported on the seismotectonic map of the Emilia-Romagna region (Boccaletti et al., 2004 and 2011; Martelli et al., 2017). We imported in Geomodeller the following data:

–   a high-resolution Digital Terrain Model at a gridsize of 10 m, provided by the Regione Emilia-Romagna Technical Office (*DTM LiDAR, Ministero dell'Ambiente e della Tutela del Territorio e del Mare*) as raster image;

–   an excerpt of the seismotectonic map of the Emilia-Romagna region in scale 1:250,000 (published by Boccaletti et al., 2004), which reports the main geological units outcropping in the area, as well as the active (and potentially active) tectonic structures;

–   two deep geological sections in scale 1:250,000 (published by Boccaletti et al., 2004) constrained by boreholes data and derived by interpreting reflection seismic profiles acquired in the area, which cross the study area in the NNE-SSW direction, transversally to the Apennine chain axis (traces AA' and BB' in Fig. 1 and Fig. 2);

–   a deep geological section in scale 1:250,000 (published by Boccaletti et al., 2011) which crosses the study area in the WNW-ESE and W-E direction, longitudinally to the Apennine chain axis (trace HH' in Fig. 1 and Fig. 2);

–   isobaths of the plain deposits bottom (Formation 'a' with age 0.45 My in Fig. 3a); (Martelli et al., 2017)

–   isobaths of the Pliocene sediments bottom (Formation 'M-P1' with age 6.3 My in Fig. 3b); (CNR, 1992)

To constrain the geometry of the geological bodies, we manually digitized the interfaces separating the oldest geological formations, both on the excerpt of the seismotectonic map and on three mentioned geological cross-sections, and merged them with the digital data outlining the isobaths of the two youngest formations bottoms. Similarly, we digitized the fault traces on the cross sections and attributed them their extension, their relationship with the geology series (in order to take in consideration the faults when interpolating the geology series) and their relationship with other faults (to define the termination of one fault on another). The building process consists in several steps, with a progressive integration of the available data, generally starting from the top surface. At each step we perform a computation of the implicit surfaces of the formation boundaries and review the partial result before adding new elements. Whether new data were available for the project, we can obtain a revised model with little effort.

The 3D model obtained for the Emilia region is displayed in Figs. 4, 5 and 6. Two different views of the model sampled on the three input geological cross-sections are shown in Fig. 4. Figure 5 shows the same two views on the entire geological volume, while Fig. 6 shows the volume corresponding to the Meso-Cenozoic carbonate succession, at left, and the fault system, at right.

In order to use the model for numerical simulations, we exported it into a 'voxet' format by sampling the geological formation volumes with a regular 3D grid.

### 2.3 Physical properties of geological formations

In order to perform the physics-based numerical simulations of the seismic waves propagation we have to assign the values of the physical properties to each 3D geological volume. Considering valid the assumption of an isotropic and visco-elastic medium, we assigned to each formation the values of the following parameters:



– the velocities $V_P$ and $V_S$ of the compressional and the shear seismic wave, respectively

– the mass density $\rho$

– the elastic quality factors $Q_K$ and $Q_\mu$ for bulk and shear deformations, respectively.

We assumed that each geological formation belonging to the stratigraphic pile of Fig. 4 is characterized by different values
of the above-mentioned parameters. In order to simplify the assignment of the physical properties values to each formation, we decided to characterize each unit only by $V_P$ and to evaluate from the latter the other four properties using some well-established empirical relations.

Considering the velocities expressed in km/s and the density in in g/cm$^3$, we adopted the relation:

$$V_S = 0.7858 - 1.2344 \cdot V_P + 0.7949 \cdot V_P^2 - 0.1238 \cdot V_P^3 + 0.0054 \cdot V_P^4 \tag{1}$$

found by Brocher (2005) for $V_S$ and the relation:

$$\rho = 1.74 \cdot V_P^{0.25} \tag{2}$$

found by Gardner et al. (1974) for the mass density.

The intrinsic attenuation is described with the shear quality factor, which is evaluated from $V_S$ with the widely used rule of thumb (e.g. Paolucci et al., 2015)

$$Q_\mu = 100 \cdot V_S \tag{3}$$

and with the bulk quality factor, which value is set as

$$Q_K = 3.5 \cdot Q_\mu \tag{4}$$

in accordance with the theory exposed by Morozov (2015).

We assumed that the value of $V_P$ assigned to each geological formation might depend on the depth through a linear gradient

$$V_P(z) = V_P(0) + \partial_z V_P \cdot z \tag{5}$$

The values of the coefficients $V_P(0)$ and $\partial_z V_P$ for each formation are given in Table (1). From Table (1) appears that in most formations a constant value for $V_P$ is assumed. The $V_P$ value in the formation A has been set to 1.5 km/s which corresponds to the velocity of the compressional seismic waves in water saturated soils. The values in the deeper formations were chosen in accordance with $V_P$ values of the geological formations in the Po Valley basin published by Montone and Mariucci
25   (2015).

### 3 Computation of seismic waves

The computation of seismic wave propagation in alluvial basins at frequencies of engineering interest represents a demanding task. The geometrical complexity requires the adoption of numerical computational methods for the solution of the viscoelastodynamic equation, which governs the ground motion during an earthquake. The wide range of wave velocities
involved in realistic simulations imposes a fine sampling of the spatial and temporal domains. The computational cost of typical applications dictates the usage of parallel algorithms suitable for exploiting HPC resources.

### 3.1 The FPSM3D code

In order to compute the seismic waves propagation in the constructed 3D geological model we adopted the code FPSM3D (Klin et al., 2010), which is based on the Fourier pseudo-spectral method (FPSM) for the integration of hyperbolic equations.



The peculiarity of FPSM (first introduced by Kreiss and Oliger, 1972) consists in the evaluation of the spatial derivatives by means of a multiplication in the wavenumber domain. The transition from the spatial domain to the wavenumber domain, and back, is performed by means of the Fast Fourier Transform. FPSM combines the optimal accuracy of the global spectral differential operators and the simplicity of the spatial discretization with a structured rectangular grid. According to the

Nyquist's sampling theorem, FPSM works with a relatively coarse spatial sampling (Fornberg, 1987), which represents a valuable advantage when solving 3D problems. The code FPSM3D performs the time integration by means of the 2nd-order explicit Finite-Difference (FD) scheme and adopts the Convolutional Perfectly Matching Layer (C-PML) approach (Komatitsch and Martin, 2007) to prevent the effects of the spatial domain boundaries on the computed wavefield. The effects of the staircase approximation of the material interfaces in the regular grid are avoided using the volume harmonic

averaging of the elastic moduli and volume arithmetic averaging of the mass density, as proposed by Moczo et al., 2002. The adequateness of the FPSM3D code in this kind of applications is demonstrated in the works by Chaljub et al. (2015) and Maufroy et al. (2015), aimed to estimate the accuracy of a number of numerical methods currently used for physics-based predictions of earthquake ground motion in 3D models of sedimentary basins.

### 3.2 Setup for the computations

In the present work we perform computations of the seismic wavefield in the frequency range 0-2 Hz. We have chosen of the maximum frequency ($f_{max}$) according to the detail of the 3D geological model. The most superficial structural unit (i.e. formation 'A') presents a variable thickness $H > 50$ m on a large part of the studied area and in particular at all the station locations. Considering that the average shear wave velocity assigned to this unit is about $Vs = 330$ m/s, the fundamental resonance frequency ($f_0$) of the upper layer results below 1.65 Hz, if we apply the known relation $f_0 = Vs/4H$. In order to

model the effects of the upper layer on the wavefield we have to set $f_{max} > f_0$. On the other hand, the lack of detail in the shallower part makes the model unsuitable for realistic computations at frequencies much higher than $f_0$, thus we set $f_{max} = 2$ Hz.

The numerical computations were performed using the spatial and temporal sampling as exposed in Table (2). The spatial domain consisted in a box with 61.4 km wide square basis and height 22 km. The vertical sampling of the spatial grid was

shrunk towards the top surface in order to sample accurately the smaller wavelengths that characterize the seismic wave-field there. The flat topography of the studied area allowed us to neglect possible topographic effects. The Courant stability criterion dictated a time sampling step as short as 0.005 s, and 65 s long time series of the ground motion were extracted in all the grid points at surface and on the two East-West and North-South vertical sections crossing the epicenter of the simulated events. The computational cost of each simulation was about 50,000 core-hours on the IBM-BG/Q supercomputer

at CINECA.

### 4 Comparisons between numerical predictions and data

In order to investigate whether the ER3D model is capable to reproduce the peculiar features of the observed earthquake ground motion, we performed a comparison between the ground motion recorded by 29 seismological stations deployed in the study area during the 2012 seismic sequence (see Fig. 1 and Table 4) and the numerically predicted ground motion at the

same locations. The considered seismic stations belong to the Italian Strong Motion Network (IT) managed by DPC, and to the Italian National Seismic Network (IV) managed by INGV. For reference, we considered also the physics-based numerical predictions resulting from the simplified model PADANIA (Malagnini et al., 2012), which is composed of horizontal homogeneous layers (therefore a 1D model in contrast to our 3D). The numerical simulations regarding the PADANIA model were performed using the Wavenumber Integration Method (WIM) (Herrmann and Wang, 1985;

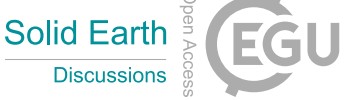

Herrmann, 1996a, b), which solves the wave equation in a horizontally layered medium. The synthetic seismograms contain all the phases and are accurate in both the near- and far-field.

### 4.1 The simulated earthquake ground motion

In order to focus the study on the propagation of seismic waves rather than on their generation, we decided not to consider

the main shocks of the 2012 Emilia sequence, which were simulated in previous works concerning the 3D modeling of the Po Plain (e.g. Molinari et al., 2015; Paolucci et al., 2015). For those events, of magnitude $M_W$ 6.1 (on 20/5/2012) and $M_W$ 5.9 (on 29/5/2012) respectively, the effects of the peculiarities of the seismic source in the recorded waveforms are not negligible. Since the main topic of this work is the estimation of the wave propagation effects on the earthquake ground motion (in particular the influence the Po Plain underground geological structure has on the wave propagation), we decided

to simulate events of lower magnitude. With an upper frequency limit of 2 Hz (see section 3.2), we can roughly assume that t the complexities (i.e. unpredictable irregularities in the spatial extension and time evolution) in the seismic sources are negligible for earthquakes up to $M_W = 4.0$. Nevertheless, such events are strong enough to be well recorded in almost all the considered stations. We therefore computed the seismic wavefield for the two $M_W \approx 4.0$ events listed in Table 3 with sources located at the NE and NW ends of the studied area (c and d labeled events in Fig. 1). The hypocenters and the magnitude of

these events were taken from the latest relocation study (Lavecchia et al., 2015). The generation of the wavefield was modelled as a double-couple point source with a time function corresponding to the low-pass minimum phase Butterworth filter plotted in Fig. 7 and with an inverse focal mechanism, in accordance to the fault plane solutions of the 2012 sequence found by Saraò and Peruzza (2012).

### 4.2 Comparison with the empirical earthquake ground motion

The permanent and temporary seismological stations deployed in the study area during 2012 are mapped in Fig. 1 and listed in Table 4. The time series recorded at these stations during the two events listed in (Table 3) are available from the Italian strong motion database ITACA (http://itaca.mi.ingv.it), Pacor et al., 2011). Events epicenters and stations locations are shown in Fig. 1. Since the empirical time-series have a much wider frequency content than the simulated ones, they were low-pass filtered using the same minimum phase Butterworth filter plotted in Fig. 7 that was used as the source time function

in the numerical simulations.

We compared the simulated ground motion with the empirical one in terms of horizontal peak ground velocity (PGV) defined as the peak modulus of the vector sum of the two horizontal components and in the duration defined as the time interval length between 5% and 95% of the Arias intensity (Arias, 1970).

In Fig. 8 we plot – separately for the two events – the logarithm of the measured and computed PGV at each station against

the epicentral distance. We represent there also the linear fit for the three series of data (empirical, 3D model ER3D, 1D model PADANIA). The plot shows that, in both cases, the 3D model numerical predictions fit better the observations, whereas the 1D model prediction underestimate the observed PGV at most stations (by a factor of almost 2). The high variability shown by stations at similar epicentral distance is probably due to the different source-station azimuth and focal mechanism-radiation pattern.

Similarly, in Fig. 9 we plot the duration of the measured and computed ground motions against the epicentral distance. Again, the 3D model numerical predictions fit better the observations than the 1D model predictions, which underestimate the duration at almost all the stations. In particular, it can be observed how the 3D model is able to predict quite well the very long duration values observed at some stations located in the southern part of the model (for example the MDN, MODE and ZPP stations for the event labelled as "d" in Fig. 1). In order to analyze the reasons of the exceptional length of the observed

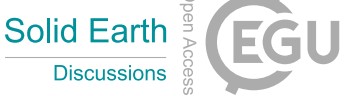



ground motion, we analyze, in the following section, the snapshots of the wave field propagation across a North-South vertical section that encompass both the source of the event "d" and the neighborhoods of the station MODE.

### 4.3 Wave propagation across a vertical profile

A remarkable advantage of 3D numerical modeling consists in the possibility to visualize the wave phenomena, which causes unexpected features in the observed ground motion. The most apparent anomalies in the observed ground motion that were reasonably well predicted with the 3D model are the PGV and the long duration at the stations south of the epicenter during the $M_W$ 4.1 12/6/2012 event (labeled with d in Fig. 1). In Fig. 10, we compare the horizontal components of the empirical and the computed time series at the stations T828, T824 and MODE. Even though we could not reach a match between the time series in a strict sense, the results obtained with the 3D model represent a significant improvement if compared to the 1D model results.

In order to investigate the cause of the particular features of the ground motion in these stations we can follow the modelled propagation of the seismic waves on a vertical profile extracted from the 3D spatial domain (Figs. 11, 12 and 13), whose trace on the surface is shown by the red dashed line in Fig. 1. The profile cuts the volume in the South-North direction and includes the $M_W$ 4.1 12/6/2012 event source (labelled with d in Fig. 1) in the northern part of the section as well as the neighborhood of the T828, T824 and MODE stations (represented with green triangles) in the central and the southern part. The grey shadow on the profiles represents the wave amplitude whereas the yellow lines represent the interfaces between the structural units. For the sake of clearness, the structural units are labeled in Fig. 11a only. The profile samples three different areas of the geological structure, the northern, the central and the southern. The northern area is characterized by the Ferrara Folds (Fig. 1), where high velocity layers ($V_S$ =1.7 km/s) are lifted up to few tens of meters below the surface. The central area is characterized by a deep syncline with thick, low velocity ($V_S <$ 1 km/s) superficial layers of sediments and alluvial deposits. In the southern part, we find the Emilia folds, which again reduce the thickness of the soil cover. In the first snapshot, taken after 2 s of propagation (Fig. 11a), we can see the initially concentric wavefronts propagating from the source located in the Ferrara folds. After 4 s (Fig. 11b) the wavefronts propagating towards south assume an almost plane shape, after having been deformed in the slower formations of the basin. We can clearly discern the compressional waves (denoted by the letter P), and the shear waves (denoted by the letter S), being the latter stronger and slower, with a shorter wavelength. After 8 seconds (Fig. 12a) the direct S has reached T828 and we can observe how at that time the S wave is reflected from the soil surface above the ridge and channeled in the dipping layers south of it. Because of the layers dip, the reflected S wave hits the layers at a post-critical angle and generates a number of diffracted waves, which correspond to surface waves overtones, if we adopt a mathematically more elegant formalism. After 16 seconds (Fig. 12b), the aforementioned diffracted waves can be well recognized in the profile across the wave-field and we can associate them with the strong phases following the direct S arrival at the 3 considered stations.

For example, the strong wave-train predicted at T824 between 16 and 20 s of propagation (Fig. 10b), corresponds to the refracted wave on the interface between the layers P and MP. The subsequent wavetrains at about 23 and 28 s correspond to the refraction on the Qm-P and B-Qm interfaces, respectively, as it appears from the snapshot at 32 s (13a). The refraction on these three interfaces originates also the three most evident wavetrains at the end of the signal at MODE, as can be understood from Figs. 13a and 13b.

The lack of a stricter match between the predicted and the observed wave-trains can be ascribed to the uncertainties in the layer geometries and physical properties and does not affect the explanation we provided here for the elongation of the ground motion in the stations south of the $M_W$ 4.1 12/6/2012 epicenter.



## 5 Conclusions

The study attests the importance of considering possible 3D heterogeneities in the geological structure in the estimation of the expected earthquake ground motion. The test case consists in the well-documented 2012 seismic crisis in Emilia-Romagna (Italy), in the middle of the Po Plain. The alluvial valley of the Po Plain presents a complex geological architecture, which may locally cause an aggravation of the ground motion during an earthquake. In order to explain the ground motion observed during some earthquakes of the 2012 Emilia seismic crisis, characterized by unexpectedly long duration and large peak ground velocity (PGV), we developed a 3D digital geological model of a limited area (a square with 70 km long side) of the Po Valley basin, by considering already published geological and geophysical data. We applied physics-based 3D numerical modeling to predict *a posteriori* the anomalous ground velocity duration and peak values from the developed model, finding a good correspondence. On the contrary, the prediction performed on the basis of a simplified model consisting of horizontal flat layers significantly underestimates these parameters. From the snapshots of the numerically evaluated seismic wave-field we could understand that the elongation of the ground motion is due to surface wave overtones originated by the post critical up-ward reflection on the sloping interfaces of the uppermost structural units of the S-wave reflected from the surface above the top of the ridge generated by the Ferrara folds. Recently, the Rayleigh wave overtones were found responsible for the long duration of ground motion in Valley of Mexico (Cruz-Atienza et al., 2016). Here we found that areas in the Po Valley can exhibit a similar phenomenon, with the remarkable difference that the surface waves in the Valley of Mexico are excited by the basin edges whereas in the Po Valley they are generated by a buried structural ridge.

Some persisting inconsistencies between the predicted and the observed data can be attributed to local errors in the 3D model as well as to errors in the assumed source parametrization for the simulated earthquakes. Additional data from more recent and/or still ongoing studies in the area (e.g. Mirandola borehole) could allow us to improve the model. The performed tests nevertheless represent an encouraging step towards a deeper understanding of the seismic hazard in the Po Plain and in similar alluvial valleys worldwide.

### Code availability

The digital 3D geological model was set up with the commercial software GeoModeller 3.3, distributed by Intrepid Geophysics (https://www.intrepid-geophysics.com/ig/index.php?page=Home, last accessed on November 2018). The numerical simulations of seismic wave propagation were performed with HPC software developed at Istituto Nazionale di Oceanografia e di Geofisica Sperimentale (OGS) and available from the corresponding author upon request.

### Data availability

ER3D model is available in GeoModeller format from the authors upon request. The seismic recordings of the 2012 Emilia sequence events can be downloaded from the Italian Accelerometric Archive - ITACA (http://itaca.mi.ingv.it, last accessed on November 2018).

### Author contributions

L. Martelli, who selected the relevant geological data for the model construction and by E. Priolo, who coordinated the project activities, conceived the work. G. Laurenzano and M. A. Romano assembled the digital 3D geological model, whereas P. Klin performed the numerical predictions.





**Acknowledgements**

The present work was accomplished under the project "Modellazioni numeriche 3D per il calcolo del moto del suolo e della risposta sismica in Emilia-Romagna", funded by Regione Emilia-Romagna. Additional support was provided by the program HPC Training and Research for Earth Sciences (HPC-TRES) (http://www.ogs.trieste.it/en/content/hpc-training-and-research-earth-sciences-hpc-tres).

**Competing interests**

The authors declare that they have no conflict of interest.

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





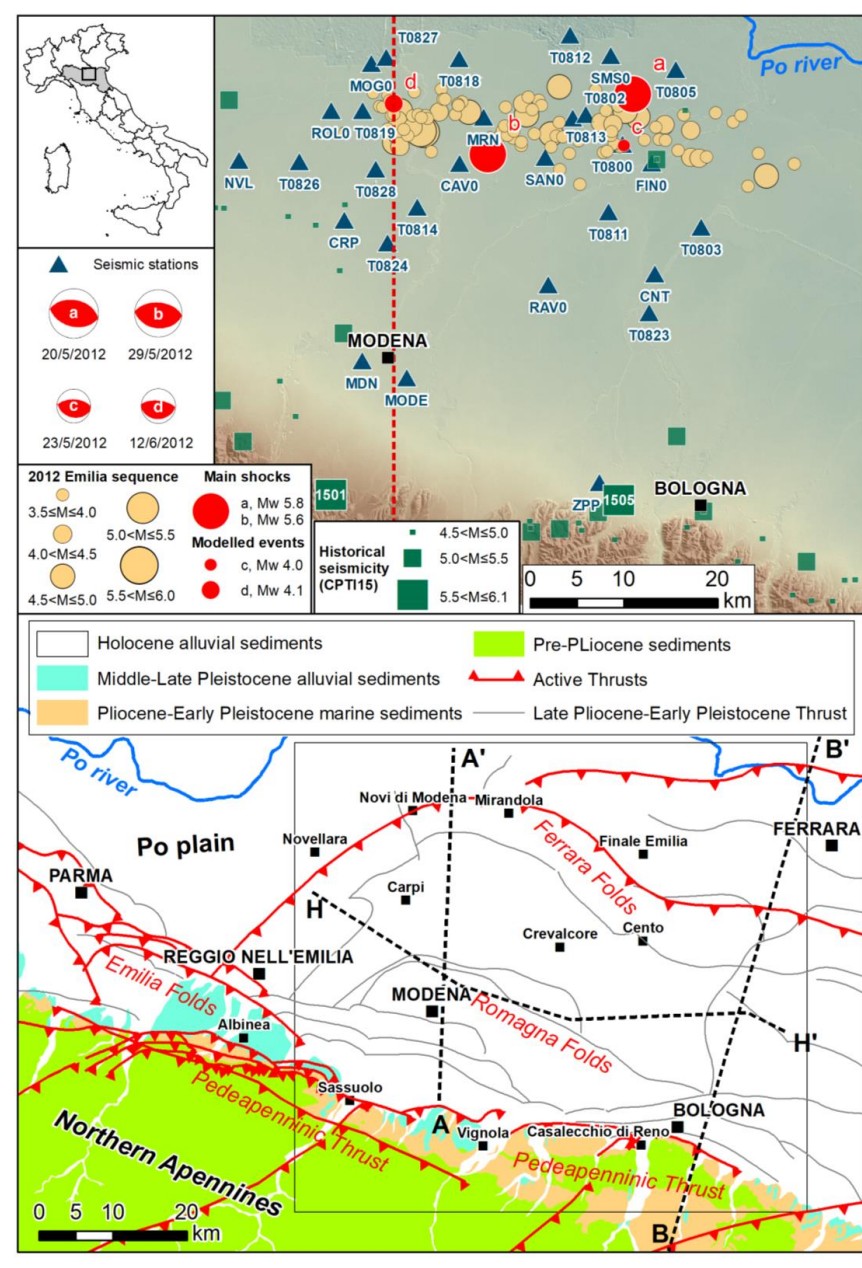

**Figure 1.** (**a**) Study area with historical seismicity (CPTI15, Rovida et al. 2016), 2012 Emilia sequence epicenters (M ≥ 3.5), temporary/permanent seismological stations and trace of vertical section of Figs. 11-13 . (**b**) Geological sketch of the study area, with traces of the three deep geological sections represented in Fig. 2.




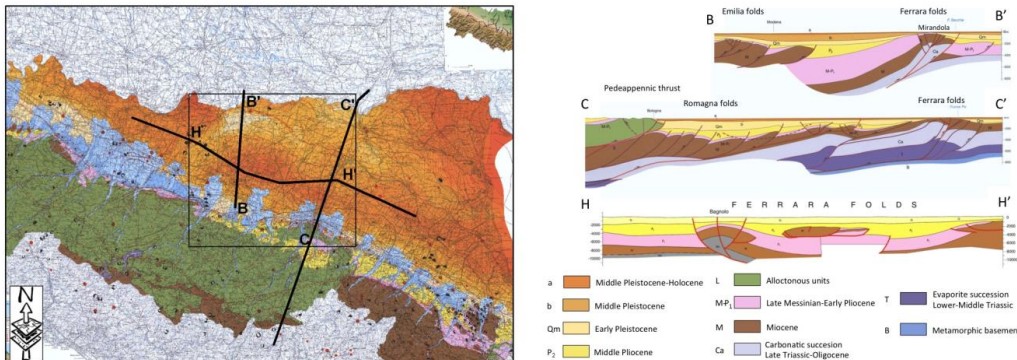

**Figure 2**. (**a**) Portion of the Seimotectonic map. Black lines represent the three geological sections traces. (**b**) Geological cross-sections from the Apenninic-Po Plain margin to the Po River (from Boccaletti et al., 2004).

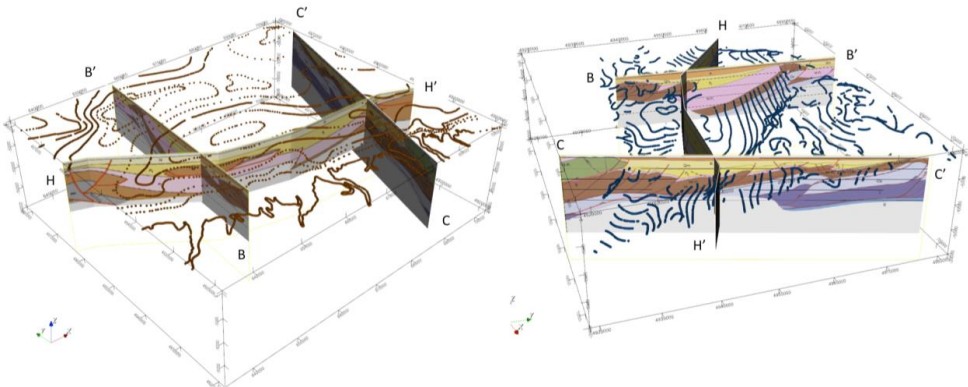

**Figure 3**. Geological cross sections and 3D surfaces. At left: South-West view and base of the plain deposits (Formation 'a'), at right: East view and base of the Late Pliocene (Formation 'M-P1').



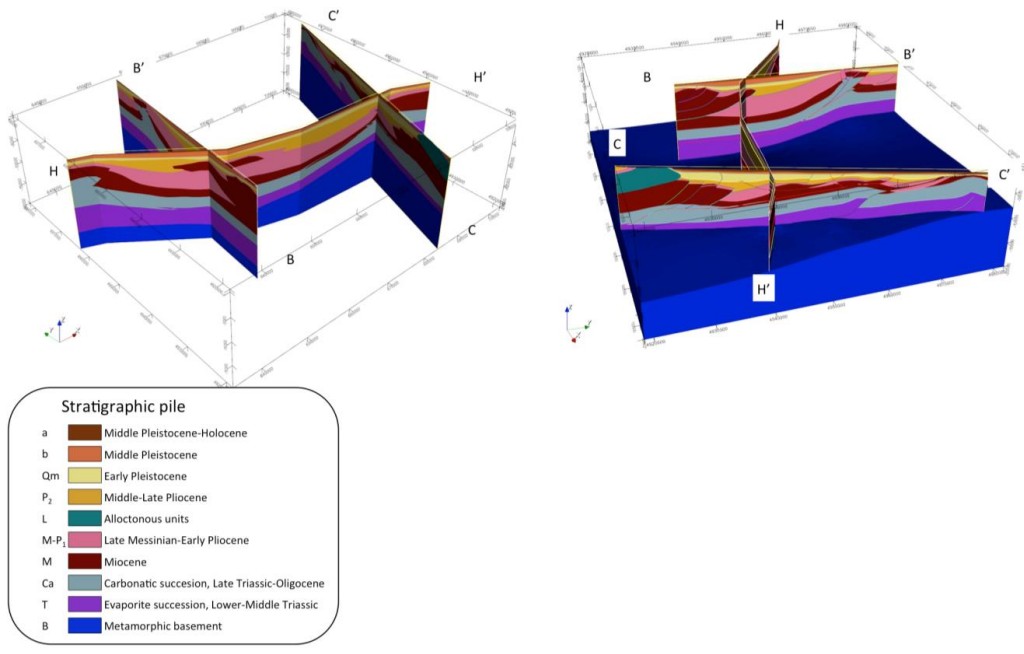

**Figure 4**. Model plotted on the three sections: South West (left) and East (right) views. The stratigraphic pile corresponds to that given in the seismotectonic map of Boccaletti et al., (2004).

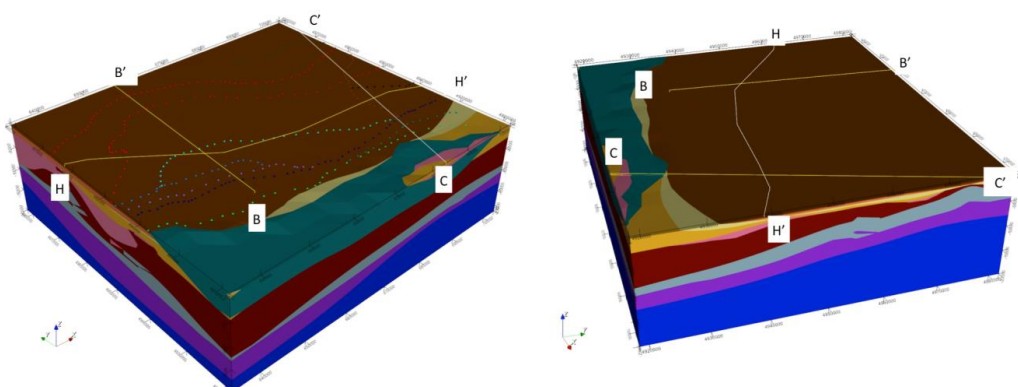

**Figure 5**. 3D model: South West (left) and East (right) views. In the South West view, the points digitalized on the seismotectonic map corresponding to the main faults are also displayed. Stratigraphic pile as in Fig. 4.





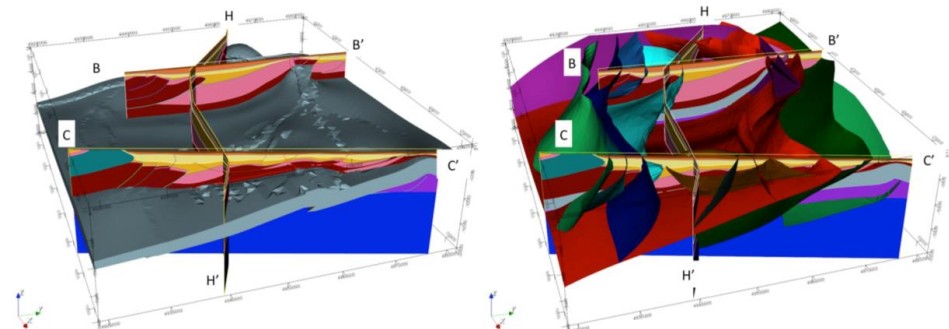

**Figure 6**. 3D model (East view). Volume corresponding to the carbonate succession (left) and fault system (right). Stratigraphic pile as in Fig. 4.

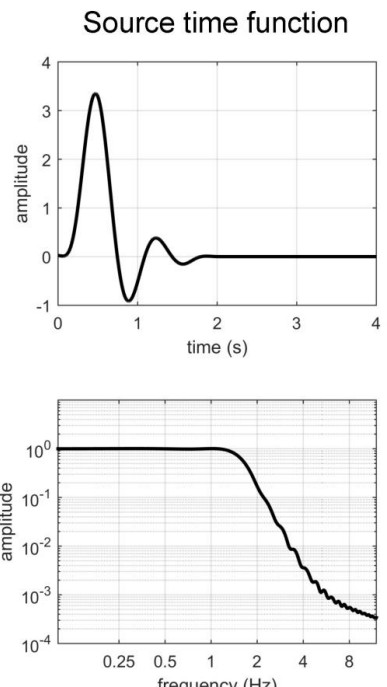

**Figure 7**. The time series and corresponding amplitude spectrum of the source time function used to excite the numerical simulation.



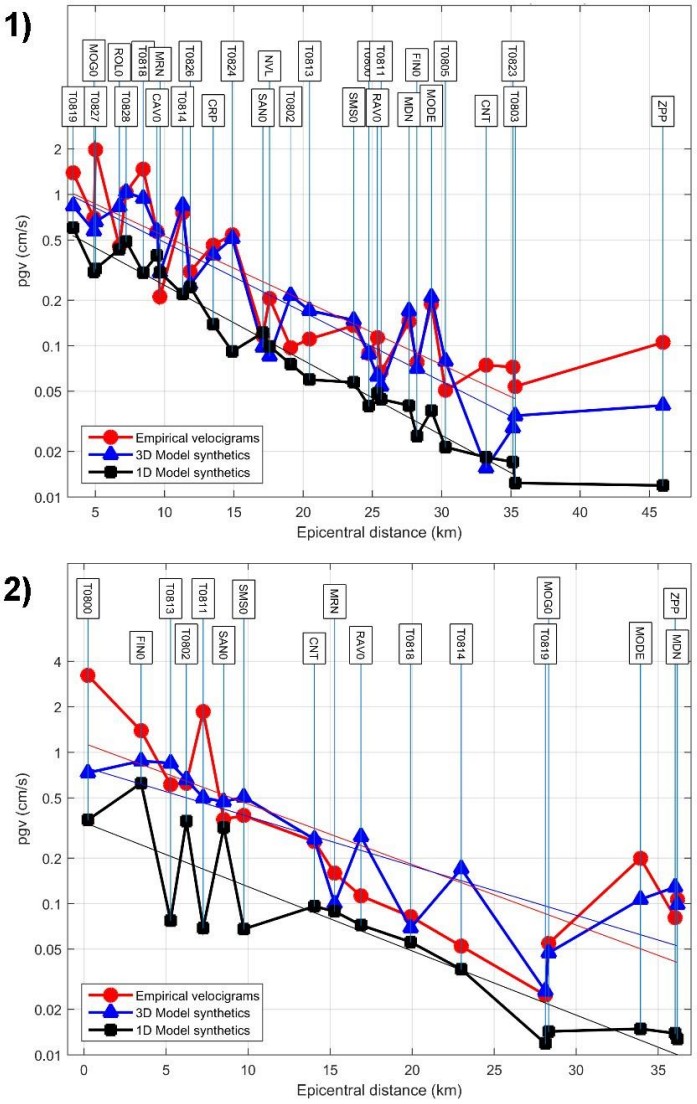

**Figure 8**. The peak ground velocity (PGV) at the considered stations plotted in function of the epicentral distance. (**1**) event $M_W$=4.1 at 01:48:36 on 2012/06/12. (**2**) event $M_W$=4.0 at 21:41:18 on 2012/05/23; The ordinate scale is logarithmic.





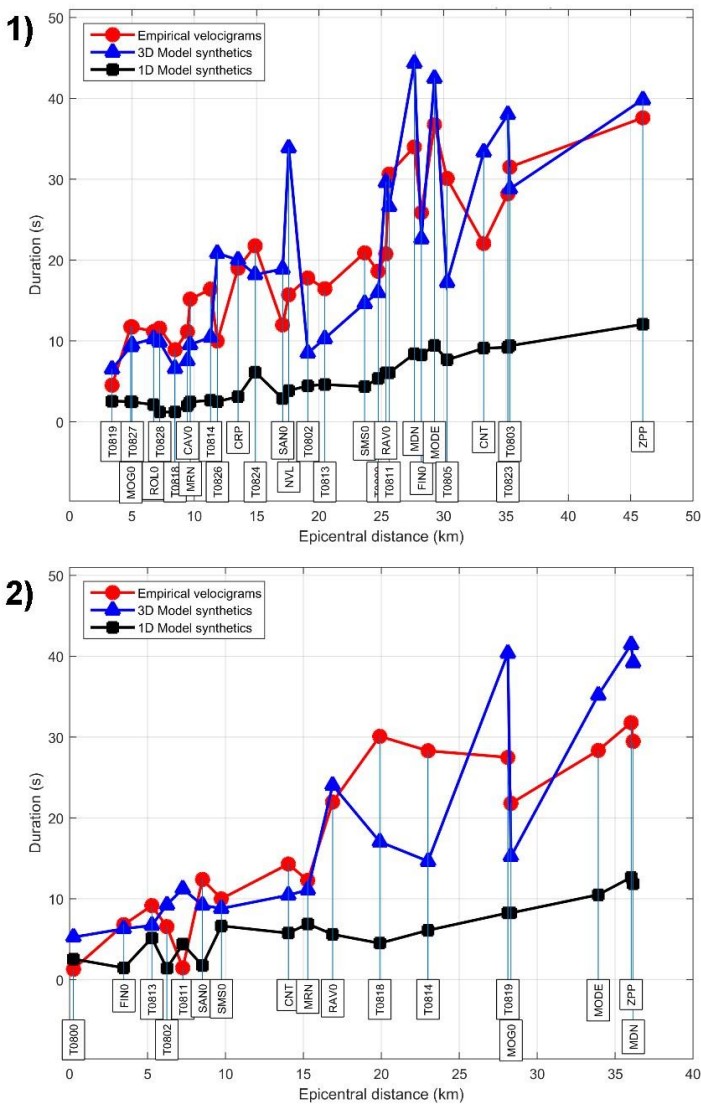

**Figure 9**. The duration (defined as interval between 5% and 95% of the Arias intensity) at the considered stations, plotted in function of the epicentral distance. (**1**) event $M_W$=4.1 at 01:48:36 on 2012/06/12. (**2**) event $M_W$=4.0 at 21:41:18 on 2012/05/23.



**Figure 10**. Comparison between measured and predicted ground velocity time series for the June 2012 $M_W = 4.1$ event in three stations southward of the epicenter. The ground motion predicted from the 3D model results significantly more consistent with the observations than the one predicted from the 1D model.



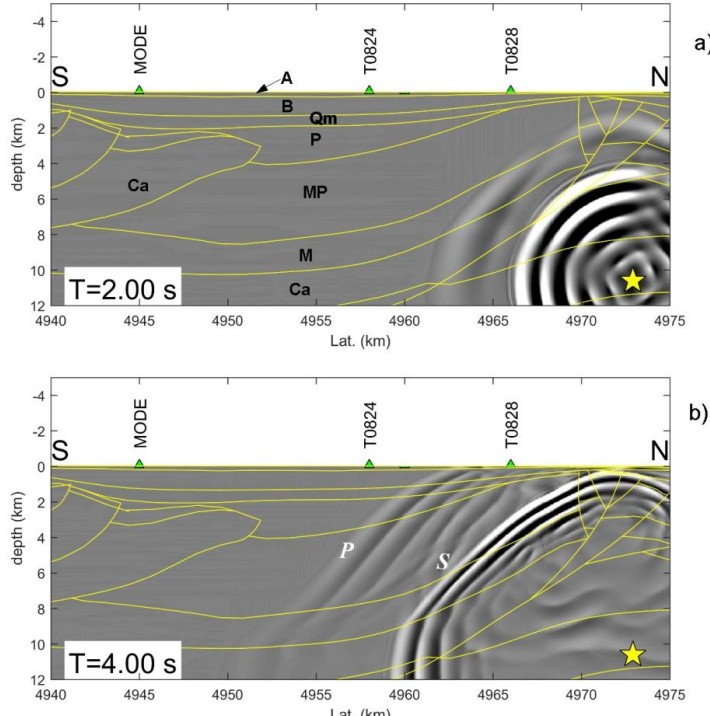

**Figure 11**. Numerically evaluated seismic wavefield for the June 2012 $M_W$ = 4.1 event across the South-North vertical section represented in Fig. 1 (red dashed line). Green triangles: projection of the nearby station locations. Yellow star: epicenter. Yellow lines: interfaces among structural units. (**a**) Snapshot taken after 2 seconds of propagation. Black letters: structural unit identifiers - see Table (1). (**b**) Snapshot taken after 4 seconds of propagation. P and S: wavefronts of the compressional and shear body waves, respectively.



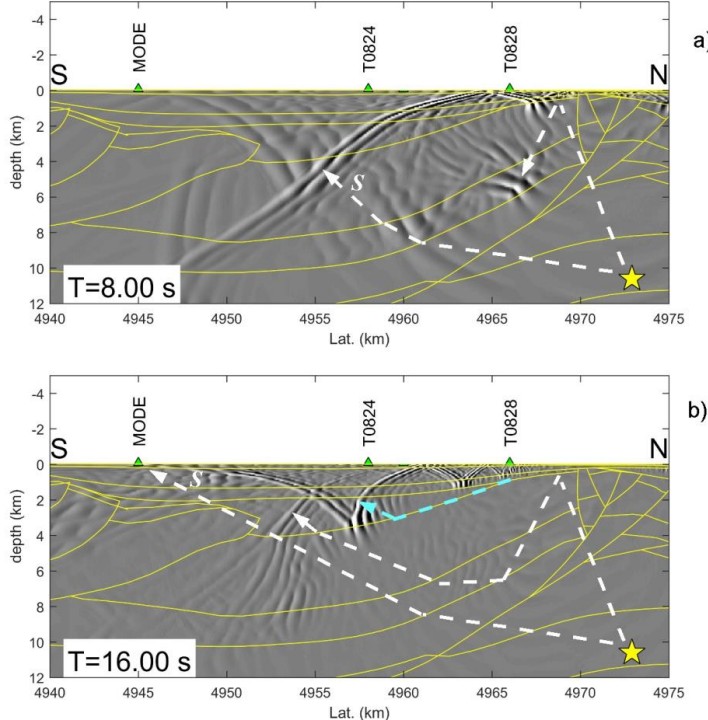

**Figure 12.** Numerically evaluated seismic wavefield for the June 2012 $M_W$ =4.1 event across the South-North vertical section. (**a**) Snapshot taken after 8 seconds of propagation. (**b**) Snapshot taken after 16 seconds of propagation. The S-wave and its refraction (evidenced by a dashed cyan line), dominate the scene.



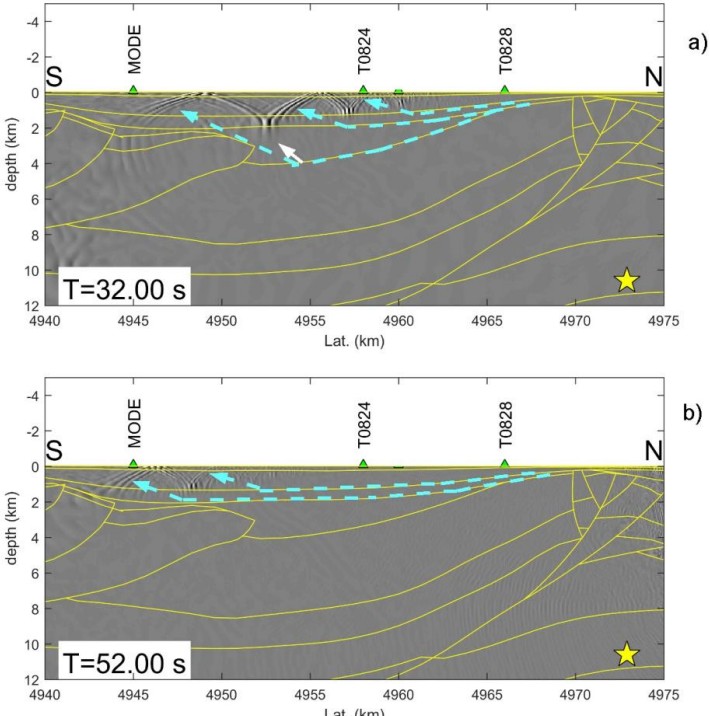

**Figure 13.** Numerically evaluated seismic wavefield for the June 2012 $M_W$ =4.1 event across the South-North vertical section. (**a**) Snapshot taken after 32 seconds of propagation. (**b**) Snapshot taken after 52 seconds of propagation. Surface waves overtones are clearly visible in the soft soil layers in the upper part are structure. The dashed cyan lines evidence the wave trains as well as the corresponding interfaces that originate the total reflection.

**Table 1.** P-wave velocity assigned to each geological formation

| Formation | Description | $V_P$ (km/s) | $\partial_z V_P$ (s-1) |
|-----------|-------------|--------------|----------------|
| A | Alluvial deposits up to .45 My | 1.5 | 0 |
| B | Middle Pleistocene sands | 1.5 | 0.5 |
| Qm | Lower Pleistocene sands | 1.6 | 0.5 |
| P | Upper-Middle Pliocene deposits | 2.6 | 0.1 |
| MP | Lower Pleistocene/Messinian marine deposits | 3.3 | 0 |
| M | Miocene Flysch | 3.4 | 0 |
| L | allochthonous Ligurides | 3.5 | 0 |
| Ca | Cenozoic and Mesozoic carbonates | 5.5 | 0 |
| T | Trias evaporites | 6.0 | 0 |
| Bas | Cristalline basement | 6.2 | 0 |



**Table 2**. Parameters defining the performed 3D numerical simulations

| | |
|---|---|
| Max. frequency | 2 Hz |
| Size of spatial grid | 1024 × 1024 × 256 |
| Grid cell dimensions (at surface) | 60 m × 60 m × 10 m |
| Grid cell dimensions (at bottom) | 60 m × 60 m × 100 m |
| Number of time integration steps | 130,000 |
| Time integration step | 0.0005 s |
| Computational cost on IBM-BG/Q | 50,000 cores |

5      **Table 3**. Parameters of the two simulated seismic events

| ID | Date (yyyy/mm/dd) | Time (UTC) | Lat(deg N) | Lon(deg E) | Depth(km) | Strike(deg) | Dip(deg) | Rake(deg) | $M_W$ |
|---|---|---|---|---|---|---|---|---|---|
| 1 | 2012/06/12 | 01:48:36 | 44.893 | 10.941 | 10.6 | 85 | 26 | 80 | 4.1 |
| 2 | 2012/05/23 | 21:41:18 | 44.847 | 11.250 | 8.9 | 105 | 33 | 101 | 4.0 |

**Table 4**. List of seismic stations

| Net code | Station code | Lon(deg E) | Lat(deg N) |
|---|---|---|---|
| IT | CAV0 | 11.0276 | 44.8343 |
| IT | CNT | 11.2867 | 44.7234 |
| IT | CRP | 10.8703 | 44.7823 |
| IT | FIN0 | 11.2867 | 44.8297 |
| IT | MDN | 10.8898 | 44.6469 |
| IT | MOG0 | 10.912 | 44.932 |
| IT | MRN | 11.0617 | 44.8782 |
| IT | NVL | 10.7305 | 44.8419 |
| IT | RAV0 | 11.1428 | 44.7157 |
| IT | ROL0 | 10.856 | 44.888 |
| IT | SAN0 | 11.143 | 44.838 |
| IT | SMS0 | 11.235 | 44.934 |
| IT | ZPP | 11.2044 | 44.5244 |
| IV | MODE | 10.9492 | 44.6297 |
| IV | T0800 | 11.2479 | 44.8486 |
| IV | T0802 | 11.1816 | 44.875 |
| IV | T0803 | 11.3508 | 44.7668 |
| IV | T0805 | 11.3226 | 44.9187 |





| IV | T0811 | 11.2265 | 44.7838 |
| IV | T0812 | 11.181 | 44.9547 |
| IV | T0813 | 11.1992 | 44.8778 |
| IV | T0814 | 10.9692 | 44.7933 |
| IV | T0818 | 11.0304 | 44.9348 |
| IV | T0819 | 10.8987 | 44.8873 |
| IV | T0823 | 11.2771 | 44.6862 |
| IV | T0824 | 10.9276 | 44.7594 |
| IV | T0826 | 10.8113 | 44.8394 |
| IV | T0827 | 10.9319 | 44.9377 |
| IV | T0828 | 10.9143 | 44.8308 |