# Peer review of "ER3D: a structural and geophysical 3D model of central Emilia-Romagna (Northern Italy) for numerical simulation of earthquake ground motion"

_Solid Earth, 2019_

## Referee Comment (RC1) · Anonymous Referee #1 · 14 Feb 2019

General

The paper by Klin et al. presents a study on the construction of a large scale 3D geological model of the Po Plain, Emilia-Romagna region (referred to as ER3D), apt for use in physics-based numerical simulations of earthquake ground motion. Special efforts have been devoted by the authors to set up the 3D subsoil model, by collecting a comprehensive set of geological and geophysical data from available studies in the literature and by integrating them in a 3D digital platform. The 3D model has been, then, used to perform 3D seismic wave propagation analyses through the Fourier pseudo-spectral method (code: FPSM3D). To validate the 3D numerical model, results

of numerical simulations were compared with the recordings obtained at the available accelerometric stations during two earthquakes of small magnitude (Mw= 4-4.1). The paper is well written and it is capable of demonstrating the superiority of 3D numerical approaches to predict peculiar features of earthquake ground motion (amplitude values and duration) in complex geo-morphological conditions like the Po Plain. I have only a few minor remarks (see list below) to be addressed before the publication in the Journal.

Remarks (1) Introduction, pg. 2, line 9-10: deviations of ground motion observations from empirical predictions are typically obtained in the near source region of large earthquakes or in complex geologic conditions (e.g. deep basins), because the GM-PEs are poorly calibrated and/or are not properly parameterized to account for those effects. Authors should specify better the reasons for the inconsistencies between GM-PEs and recordings. (2) Introduction, pg. 2, line 29-30: with reference to Paolucci et al. (2015), the satisfactory agreement between simulated and recorded motions was not only due to the modelling of the extended seismic source but also to that of the most significant geologic discontinuities. The latter were demonstrated to be critical to explain the propagation of surface waves towards North and South. (3) Four references are missing: Guillen et al., 2004; Lajaunie et al., 1997; Chiles et al., 2004; Chiles et al., 2006. (4) Section 2.3: I suggest the authors to improve the description on how the elastodynamic properties of the soils were defined. Authors provide the basic relationships between Vp-Vs, Qs-Vs and Vp as a function of depth, but further details on how these functions were calibrated should be provided (which data? References?). Furthermore, in Table 1 values of Vs and Qs should be also provided besides Vp and its gradient, as they are fundamental (more than Vp) for any site response model. How does the Vs velocity model proposed by the authors compare with the ones available from geophysical surveys in the (e.g. Milana et al. 2014)?

Milana, G., Bordoni, P., Cara, F., Di Giulio, G., Hailemikael, S.&Rovelli, A., 2014. 1D velocity structure of the Po River plain (Northern Italy) assessed by combining strong

motion and ambient noise data, Bull. Earth. Eng., 12, 2195–2209

(5) Section 4.3: referring to Fig. 8, are the horizontal PGV values? Geometric mean or maximum of horizontal components of ground motion? What about vertical component? From Fig. 8, it is noted that at very short epicentral distances, typically less than 5 km, PGV from recordings are higher than the simulated ones, for both events. Furthermore, I encourage the authors to extend the comparison between recordings and synthetics by showing for selected stations a clearer comparison in terms of velocity waveforms and corresponding Fourier amplitude spectra (at least for some components of ground motion).

Editorial typos - Pg. 3, line 24: Boccalletti et al. 20111 - Eq. (3): specify that Vs is expressed in km/s - Pg. 7, line 10: Moczo et al. (2002) - Pg. 8, line 10: remove "t" at the end of the line - Table 2: add rows for Vs and Qs, as commented above, and change Vp to Vp(z=0).
* * *

---

## Referee Comment (RC2) · Anonymous Referee #2 · 27 Feb 2019

The manuscript describe the construction of a 3D geophysical model of a 60km x 60km x 20km volume in the Po Plain basin (Italy) using some informations on the interfaces geometry and elastic properties available in the literature. The main aim of the 3D model is the simulation of seismic wavefield in the region. The authors implemented the model in a 3D computational mesh for a Fourier pseudo-spectral element method. They claim that the final model and the computational mesh is able to accurately simulate frequency up to 2 Hz. They compare the simulated seismograms with the recorded ones, comparing the PGV and the duration (Airy duration) for two events recorded by ∼ 29 stations. The main conclusion of the work is that the 3D model is able to reproduce the long shaking duration observed in the data, better than

a 1D model. The authors also analyse a 2D section of the simulated wavefield in order to understand from which part of the model the wiggles observed many seconds after the origin time come from.

As general consideration, the paper appears well written, the topic might be of interest of the journal and I would suggest it for publication in Solid Earth after a revision following the remarks below.

General comments: The method and the aim of the manuscript is not original per se. As mentioned by the authors, there are already 3 published work in which, with different aims, region and methods, the authors built a 3D model of the basin for seismic wave propagation purposes. This would be the 4th attempt. For that reason, I strongly suggests the authors to make their model publicly available (downloadable by the public) and if it possible also the a priori data used, in order to allow a comparison between the response of the various models and the reproducibility of the final result.

From some of the papers already published, the conclusion that a 3D model is better than a 1D model was already clear. It would be interesting to highlight here what is better or different for this model.

I am a bit skeptical regarding the accuracy of 2Hz in the simulations, especially because the 3D model does not have such detail in the data used to built the model itself. Some more word regarding this point will be appreciated. In particular it is not so clear how it is possible to reach such details in the model if only 3 2D sections (+ 2 interfaces) are used as input for the 3D model constructions (more comments on this point are below).

And other consideration is the following: in this work I do not see a real quantitatively comparison or validation of the 3D model. The authors only use two events and compare only PGV and Arias duration (without showing any example). The full waveform is not really compared.

Comments (following the text):

P2/L1-3: This period is a bit too long. What do you mean for geophysical model?

P2/L18: developed—> built?

P3/L5-7: Is it 2Hz too optimistic?

P3/L21: In order to set up of a reliable —> delete "of"

P3/L25: Boccaletti et al. 20111 —> check the year

P4/L31-41: The authors use a commercial software to built the 3D model. The procedure of building the model is not so clear (it appears as a black box), especially how the software creates the features and the interfaces between the three 2D sections and the two horizontal horizon (Pliocene deposit and plane deposit), where no information is available. I suggest to improve this part.

P5/L18-19: what is new in the 2D sections from Boccaletti et al., 2004 and 2011, if compared with Pieri and Groppi (1981)? Better interpretations of the original data?

Figure 4-5-6: It would be nice to show a section of the 3D model that so not follow the 2D section used as input data, in order to show the final result in the region where the model is really created. I have noticed that, in figures 11-12 one 13, the authors show a section that do not coincide with the ones used to build the model. However in this section, it is possible to notice that there are some very small details at the section borders (up-left and up-right) not labeled with any structural units. What are they? Horizons? Faults? Which structural unit they belong with? How do you know these are real features? The 3 figures are not at a high resolution. Labels are not readable.

P6: The assignment of the physical properties is pretty standard. Why do not include more local informations? In Table 1, I would show also the VS and density values for each geological formation.

P7-L15: frequency rage: 0-2 Hz. Really from 0 Hz? In A computational mesh of 60km

x 60km x 20km you cannot simulate long wavelength.

P7/L27: The simulated signals are 65s long. Could it be that you are missing some late arrival (at time > than 1 minute after the OT)? Did you test this point?

P8/L26-28: I suggest to show in one figure an example, for one event and one or two stations, of the 3 compared signals (data, 3D, and 1D model) and the measure done on these signals (Arias duration , marking the 5% and 95% and PGV). I do not find very satisfactory the duration boxes showed in figure 10.

Figure 10: Why you do not show the Z component? And instead you show the Norm? "Empirical" are the Data, right? Of course we do not expect that the 3D model is able to match every wiggle of the data; however comparing data and synthetics from 3D model, it is clear that the arrival time and the number of energy packages in the data are not the same as the simulated ones. This might be mainly due, in my opinion, to wrong interfaces in the 3D model that causes or not, spurious arrivals. Also the first arrival appears to not fit the data. Can the author comment on that? Moreover the duration boxes plotted in the figures appear to have a strange starting and ending point, neglecting some shaking.

Figure 11-12-13: I would merge the 3 figures in a single panel, if possible.

Figure 12, panel b): what is the wave package that appear to propagate from bottom-left to up-right, near the station T0824 (at its left), at the section top?

In the text sometimes the authors write "elongation of the ground motion". I would change it with long shaking duration or something like that.

It would be great, for the geophysical community, to have the model publicly available, in order to allow other scientists to compare models and to add, if possible new data.

---

## Author Comment (AC1) · 27 Mar 2019

The authors kindly thank the anonymous Reviewer for the comments on their manuscript. In the following the authors provide the answer to each remark and illustrate the corresponding changes that were applied to the manuscript.

Reviewer's remark (1)

Introduction, pg. 2, line 9-10: deviations of ground motion observations from empirical predictions are typically obtained in the near source region of large earthquakes or in complex geologic conditions (e.g. deep basins), because the GMPEs are poorly

calibrated and/or are not properly parameterized to account for those effects. Authors should specify better the reasons for the inconsistencies between GMPEs and recordings.

Authors' answer:

We agree on specifying better the reasons for the inconsistencies. Therefore we substituted the sentence "Those deviations are usually due to physical phenomena that in principle can be taken into account by using the numerical-deterministic method. " on pg.2, line 10 with the following: "Those deviations imply the presence of case-specific features in wave generation or propagation (e.g., complex fault ruptures, complex geological structures, such as deep basins), which are not adequately considered in the derivation of the GMPE. In order to predict the effects of these features we may apply numerical-deterministic methods".

Reviewer's remark (2)

Introduction, pg. 2, line 29-30: with reference to Paolucci et al. (2015), the satisfactory agreement between simulated and recorded motions was not only due to the modelling of the extended seismic source but also to that of the most significant geologic discontinuities. The latter were demonstrated to be critical to explain the propagation of surface waves towards North and South.

Authors' answer:

The reviewer is right, therefore we substituted the sentence "The overall satisfactory agreement of their simulated waveforms with the empirical records was however attributed principally to the assumed extended source model (i.e. slip distribution and rupture propagation) rather than to their model structure, which contains only two main geologic interfaces." with the following one: "The overall satisfactory agreement of their simulated waveforms with the empirical records is due to two key-elements: the extended source model (i. e. slip distribution and rupture propagation) and the 3D structural model, which contains only two main geologic interfaces, (i. e. the base of the Pliocene formation and that of the Quaternary deposits). In particular, the satisfactory simulation of the surface waves trains stems mainly from the shape of the interface of the base of the Quaternary deposits."

Reviewer's remark (3)

Four references are missing: Guillen et al., 2004; Lajaunie et al., 1997; Chiles et al., 2004; Chiles et al., 2006.

Authors' answer: We removed the erroneous citation Chiles et al., 2006 and added the following items to the list of references:

Chiles, J.P., Aug, C., Guillen, A., and Lees, T., 2004, Modelling the Geometry of Geological Units and its Uncertainty in 3D From Structural Data: The Potential-Field Method: Proceedings of "Orebody Modelling and Strategic Mine Planning", Perth, WA, 22 - 24 November 2004, AusIMM, 313-320.

Guillen, A., Courrioux, G., Calcagno, P., Lane, R., Lees, T., and McInerney, P., 2004, Constrained gravity 3D litho- inversion applied to Broken Hill: Extended Abstract, ASEG 17th Geophysical Conference and Exhibition, August 2004, Sydney.

Lajaunie, C., Courrioux, G., Manuel, L., (1997): Foliation fields and 3D cartography in geology: Principles of a method based on potential interpolation. Mathematical Geology 29 (4), 571-584.

In consideration of the changes requested by the other Reviewer, the following reference was added as well: Calcagno, P., Chilès, J.-P., Courrioux, G., Guillen, A., (2008): Geological modelling from field data and geological knowledge: Part I. Modelling method coupling 3D potential-field interpolation and geological rules: Recent Advances in Computational Geodynamics: Theory, Numerics and Applications. Physics of the Earth and Planetary Interiors 171 (1-4), 147-157.

Reviewer's remark (4)

Section 2.3: I suggest the authors to improve the description on how the elastodynamic properties of the soils were defined. Authors provide the basic relationships between Vp-Vs, Qs-Vs and Vp as a function of depth, but further details on how these functions were calibrated should be provided (which data? References?). Furthermore, in Table 1 values of Vs and Qs should be also provided besides Vp and its gradient, as they are fundamental (more than Vp) for any site response model. How does the Vs velocity model proposed by the authors compare with the ones available from geophysical surveys in the (e.g. Milana et al. 2014)?

Authors' answer:

Following the Reviewer's suggestion, we improved the description of the elastodynamic properties. To this aim, we substituted the line "found by Brocher (2005) for VS and the relation: " after equation (1) with the following text:

"that was found by Brocher (2005) from a large number of measurements made in a variety of lithologies including Quaternary alluvium and Miocene sedimentary rocks, which constitute a fundamental part of our model. We also adopted the well established relation (eq.2 follows)"

At the end of the section we added the following discussion:

"We tested the validity of eq. (1) by analyzing the consistency of the predicted Vs with some measures of Vs resulting from geophysical surveys performed in the Po plain. According to eq. (1), the value Vp=1.5 km/s assigned to the uppermost formation A (table 1) - having a thickness of the order of 100m on most part of the area – turns out in Vs=0.34 km/s. This value is compatible with the average value found for Vs with ESAC method by Priolo et al. (2012) at three different sites of the Po Plain in a similar formation down to a depth of 120 m. At larger depths the proposed geological model presents significant lateral heterogeneities and could not be directly compared with the existing 1D Vs profiles that were derived from surface waves' dispersion by Malagnini et al. (2012) and Milana et al. (2014) in the frequency bands of 0.083-0.33

Hz and 0.15–0.70Hz, respectively. For example, in the depth range between 2 and 4 km our model features the simultaneous presence of very different formations, such as the Miocene and Late Messinian-Early Pliocene formations (M and MP, respectively, with Vs in the order of 1.7 km/s) and the Carbonatic succession (Ca, with Vs velocity as high as 3.3 km/s). On the other hand, the two "empirical" 1D velocity structures previously cited feature velocities between Vs=2.0 km/s and Vs=2.5 km/s within the same depth range, which are compatible with the average value of the Vs values found in our model."

We added the following items to the references:

E. Priolo, M. Romanelli, C. Barnaba, M. Mucciarelli, G. Laurenzano, L. Dall'Olio, N. Abu Zeid, R. Caputo, G. Santarato, L. Vignola, C. Lizza and P. Di Bartolomeo (2012): The Ferrara thrust earthquakes of May-June 2012: preliminary site response analysis at the sites of the OGS temporary network. Annals of Geophysics, 55, 4 ; doi: 10.4401/ag-6172

G. Milana, P. Bordoni, F. Cara, G. Di Giulio, S. Hailemikael, A. Rovelli (2014): 1D velocity structure of the Po River plain (Northern Italy) assessed by combining strong motion and ambient noise data. Bull. Earth. Eng., 12, 2195–2209.

Moreover, we added a column with Vs, Qs, density and Qk values in table 1, as suggested by both Reviewers.

Reviewer's remark (5)

Section 4.3: referring to Fig. 8, are the horizontal PGV values? Geometric mean or maximum of horizontal components of ground motion? What about vertical component? From Fig. 8, it is noted that at very short epicentral distances, typically less than 5 km, PGV from recordings are higher than the simulated ones, for both events. Furthermore, I encourage the authors to extend the comparison between recordings and synthetics by showing for selected stations a clearer comparison in terms of velocity

waveforms and corresponding Fourier amplitude spectra (at least for some components of ground motion).

Authors' answer: In order to answer to the Reviewer's remarks we made the following changes:

a) we substituted the first sentence in the caption of Fig. 8 with the following one: "Peak ground velocity (PGV, peak value of the two horizontal components) at the considered stations as a function of the epicentral distance."

b) in paragraph 4.2, on page 8, after the sentence "We compared the simulated ground motion with the empirical one in terms of horizontal peak ground velocity (PGV) defined as the peak modulus of the vector sum of the two horizontal components and in the duration defined as the time interval length between 5% and 95% of the Arias intensity (Arias, 1970)." we added the following text: "The vertical component was excluded from this comparison since it was systematically lower than the horizontal ones.".

c) in paragraph 4.2, on page 8, in the discussion regarding Fig. 8 after the final sequence "The high variability shown by stations at similar epicentral distance is probably due to the different source-station azimuth and focal mechanism-radiation pattern." we added the following sentence:

"As observed in Maufroy et al (2015), the uncertainty in source characteristics may impact the numerical predictions especially at short distances. The remarkable underestimation of PGV for the event 2 at station T800, located just above the hypocenter is therefore not too surprising and could be attributed to the combined effect of inaccurate hypocentral location, focal mechanism, and near-source heterogeneities. In fact, considering that source 2 has a dip of $33°$ (Table 2), T800 is near to the P-wave radiation maximum and at the margin of the S-wave lobe. Figure 10 confirms this interpretation: the simulated seismogram features a pronounced P-wave amplitude in the vertical component, if compared to the S-wave one. On the other hand, in the same Figure 10, the recorded seismogram presents a reversed picture: the relatively weak
P-wave (smaller than the simulated one) and strong S-wave indicate that the actual source characteristics are different from what we assumed. "

d) we changed figure 10 with a plot showing a clearer comparison in terms of waveforms, including vertical components as well as Fourier amplitude spectra. In order to support the explanation of the discrepancy in the amplitudes at station T800 we also added waveform comparisons related to the event 2.

Editorial typos

Pg. 3, line 24: Boccalletti et al. 20111 - Eq. (3): specify that Vs is expressed in km/s - Pg. 7, line 10: Moczo et al. (2002) - Pg. 8, line 10: remove "t" at the end of the line - Table 2: add rows for Vs and Qs, as commented above, and change Vp to Vp(z=0).

Authors' answer: Corrected

---

## Author Comment (AC2) · 27 Mar 2019

The authors kindly thanks the anonymous Reviewer for the remarks on their manuscript. In the following the authors provide the answer to each remark and illustrate the corresponding changes that were applied to the manuscript.

Reviewer's general comments:

The method and the aim of the manuscript is not original per se. As mentioned by the authors, there are already 3 published work in which, with different aims, region and methods, the authors built a 3D model of the basin for seismic wave propagation pur-

poses. This would be the 4th attempt. For that reason, I strongly suggests the authors to make their model publicly available (downloadable by the public) and if it possible also the a priori data used, in order to allow a comparison between the response of the various models and the reproducibility of the final result.

Authors' answer:

We agree with the Reviewer's suggestion and therefore we will make the model publicly available via https://github.com. Moreover we add as supplementary material to the present manuscript a pdf file with 3D content which allows the reader to have a more comprehensive insight to the model. On the contrary, it is not appropriate to distribute the "a priori data", because they are all already published material with references given in the manuscript.

Reviewer's remark (1)

From some of the papers already published, the conclusion that a 3D model is better than a 1D model was already clear. It would be interesting to highlight here what is better or different for this model.

Authors' answer:

In order to comply with the Reviewer's request we substituted the sentence "In the present work we focus on a more detailed 3D geological model of a limited area of the Po Plain, bounded by the Po river right bank at North, by the Northern Apennines morphological margin at South, and located between the two chief towns of Reggio Emilia at West, and Ferrara at East (Fig. 1)." in chapter 1) Introduction, line 37, with the following one:

"Among the cited works only Paolucci et al. (2015) provided the elements for understanding the peculiar features of the near-source strong-motion observed during the 2012 events (such as the propagation of prominent trains of surface waves in the Northern direction), by adopting a reasonably simple 3D model of an area centered

on the 2012 MW 6.1 mainshock epicenter. In the present work we instead focus on the southern sector of the 2012 epicentral area, characterized by a very deep basin with sediment thickness exceeding 8000 m in some points. In order to investigate the effects of this complex geological setting, we set up a 3D geological model with unprecedented detail of a limited area of the Po Plain, bounded by the Po river right bank at North, by the Northern Apennines morphological margin at South, and located between the two chief towns of Reggio Emilia at West, and Ferrara at East (Fig. 1)."

Reviewer's remark (2)

I am a bit skeptical regarding the accuracy of 2Hz in the simulations, especially because the 3D model does not have such detail in the data used to built the model itself. Some more word regarding this point will be appreciated. In particular it is not so clear how it is possible to reach such details in the model if only 3 2D sections (+ 2 interfaces) are used as input for the 3D model constructions (more comments on this point are below).

Authors' answer:

In the section "3.2 Setup for the computations", first paragraph, we provided a quantitative argument which supports our choice of 2 Hz as upper frequency limit. In the same paragraph however we explicitly mention that the model is unsuitable for realistic computations at frequencies higher than that, because of the lack of detail, as pointed out by the Reviewer. We come back on this point in the answer to the Reviewer's remark 13.

Reviewer's remark (3)

And other consideration is the following: in this work I do not see a real quantitatively comparison or validation of the 3D model. The authors only use two events and compare only PGV and Arias duration (without showing any example). The full waveform is not really compared.

Authors' answer: Actually in the manuscript we never claim that the model contains sufficient detail to reproduce accurately the full waveform. As it is expressly stated in the abstract, in the introduction and in the conclusion of the manuscript, our aim was building a model which is capable to reproduce the observed ground motions in terms of peak ground velocity and signal duration. In the conclusions we have also discussed about the possible origin of persisting inconsistencies between the predicted and the observed data. Indeed, a comparison among waveforms is shown in figure 10, where we compare the components of the predicted and the observed waveforms at three stations that appear in the snapshots along vertical profiles of figures 11-13. This was done in order to allow the reader to associate the wave-trains in the time series with the wave-fronts propagating in the subsoil structure.

Reviewer's remark (4)

P2/L1-3: This period is a bit too long. What do you mean for geophysical model?

Authors' answer: Following the Reviewer's suggestion we substituted the period with the new following sentences, where we explain also the term geophysical model as the assumed spatial distribution of visco-elastic properties in a volume of the Earth's crust.

"The present study concerns the set-up of a 3D structural model starting from geological data and the development of the corresponding geophysical model by assigning visco-elastic properties to each structural unit. The scope of the final 3D geophysical model is to allow physics-based forward modeling of seismic wave propagation aimed at 1) explaining the ground motion peculiarities observed in past earthquakes and 2) increasing the reliability of ground motion predictions for possible future events."

Reviewer's remark (5)

P2/L18: developed -> built?

Authors' answer: We substitute "developed" with "built".

Reviewer's remark (6)

P3/L5-7: Is it 2Hz too optimistic?

Authors' answer: As mentioned in the answer to the remark (2), we provided arguments for our fmax=2Hz choice in the section 3.2. To facilitate the readability, we added at the beginning of the present sentence: "As discussed in section 3.2..."

Reviewer's remark (7)

P3/L21: In order to set up of a reliable... delete "of"

Authors' answer: Corrected.

Reviewer's remark (8)

P3/L25: Boccaletti et al. 20111 -> check the year

Authors' answer: Corrected.

Reviewer's remark (9)

P4/L31-41: The authors use a commercial software to built the 3D model. The procedure of building the model is not so clear (it appears as a black box), especially how the software creates the features and the interfaces between the three 2D sections and the two horizontal horizon (Pliocene deposit and plane deposit), where no information is available. I suggest to improve this part.

Authors' answer: We agree with the Reviewer's remark and we revised the initial paragraph of the section 2.2 in a hopefully clearer form. In particular we have substituted the text

"GeoModeller is a software tool for building complex, steady state and implicit 3D geological models, directly from geological observations. The interpolation method is based on the potential field theory (Lajaunie et al., 1997; Chiles et al., 2006), so that geological interfaces (i.e. the upper or lower surfaces of the geological units) are modelled as iso-surfaces of a scalar potential field defined in the 3D space. Structural data

are treated as the gradient of the field. The interpolation of the field uses cokriging to take both contacts and structural data into account and generates surfaces that honour all the data together (McInerney et al., 2005). The adopted approach also employs rule-based modeling to control the relationships in the stratigraphic pile (either 'onlapping' or 'erosional'), and to control fault chronology within the fault network (Chiles et al., 2004). All the details about the application of this method for building geological models can be found in Calcagno et al. (2008)."

with the following one

"GeoModeller is a software tool for the integration of different geometrical, geological, and geophysical data in a geometrically coherent 3D geological model. The procedure is based on the potential field interpolation (Lajaunie et al., 1997) and is particularly well suited when the available geological data consist only in some geological maps, sparse cross-sections or boreholes. The method requires in input the location of the geology interfaces and orientation data at some points. The theory behind the method describes the 3D geologic surfaces as iso-potential surfaces of a scalar potential-field with orientation vectors playing the role of the field's gradient. The stratigraphic pile is defined by the chronological order of the strata and the relationships between the formations in terms of either 'onlap' or 'erode'. The complex geology is described by different domains, each characterized by a geological serie, separated by stratigraphic or tectonic discontinuities. For each domain, the geology is modeled by a set of sub-parallel, smoothly curving surfaces using the potential-field functions. Cokriging is used to obtain a solution that honors the input data (McInerney et al., 2005). Faults are taken into account as discontinuous drift functions into the cokriging equations (Chilès et al., 2004). Refer to Calcagno et al. (2008) for a more comprehensive description of the methods implemented in GeoModeller." The following reference was added:

Calcagno, P., Chilès, J.-P., Courrioux, G., Guillen, A., (2008): Geological modelling from field data and geological knowledge: Part I. Modelling method coupling 3D potential-field interpolation and geological rules: Recent Advances in Computational

Geodynamics: Theory, Numerics and Applications. Physics of the Earth and Planetary Interiors 171 (1-4), 147-157.

Reviewer's remark (10)

P5/L18-19: what is new in the 2D sections from Boccaletti et al., 2004 and 2011, if compared with Pieri and Groppi (1981)? Better interpretations of the original data?

Authors' answer: In order to answer to the question raised by the Reviewer, we added the following sentence at the end of P5/L18-19: "The geological cross-sections of Boccaletti et al. (2004) and Boccaletti et al. (2011) are based on more recent seismic profiles than those used by Pieri and Groppi (1981) and take into account also stratigraphic data derived from RER and ENI-Agip (1998), for the definition of the superficial part (down to a depth of approximately 300-400 m)."

We also added the following reference: RER and ENI–AGIP (1998): Riserve idriche sotterranee della Regione Emilia-Romagna. Di Dio G. (Editor). Regione Emilia-Romagna – ENI Agip, Divisione Esplorazione e Produzione. S.EL.CA., Florence, Italy (in Italian).

Reviewer's remark (11)

Figure 4-5-6: It would be nice to show a section of the 3D model that so not follow the 2D section used as input data, in order to show the final result in the region where the model is really created. I have noticed that, in figures 11-12 one 13, the authors show a section that do not coincide with the ones used to build the model. However in this section, it is possible to notice that there are some very small details at the section borders (up-left and up-right) not labeled with any structural units. What are they? Horizons? Faults? Which structural unit they belong with? How do you know these are real features? The 3 figures are not at a high resolution. Labels are not readable.

Authors' answer: We have followed the Reviewer's suggestion and substituted figure 5 with a figure showing four vertical equally spaced North-South 2D sections across

the investigated volume. We improved the labels in all figures. In figure 11 we added labels to the structural units also on at the section borders.

Reviewer's remark (12) P6: The assignment of the physical properties is pretty standard. Why do not include more local informations? In Table 1, I would show also the VS and density values for each geological formation.

Authors' answer: We thank the Reviewer for the suggestion. We added the following sentences in P6, which in part also answer to a comment of the other reviewer: "We tested the validity of eq. (1) by analyzing the consistency of the predicted Vs with some measures of Vs resulting from geophysical surveys performed in the Po plain. According to eq. (1), the value Vp=1.5 km/s assigned to the uppermost formation A (table 1) - having a thickness of the order of 100m on most part of the area – turns out in Vs=0.34 km/s. This value is compatible with the average value found for Vs with ESAC method by Priolo et al. (2012) at three different sites of the Po Plain in a similar formation down to a depth of 120 m." As suggested by both Reviewers, we added a column with Vs, Qs, density and Qk values in table 1.

Reviewer's remark (13) P7-L15: frequency rage: 0-2 Hz. Really from 0 Hz? In A computational mesh of 60kmx 60km x 20km you cannot simulate long wavelength.

Authors' answer: The reviewer is right therefore we have substituted the sentence "In the present work we perform computations of the seismic wavefield in the frequency range 0-2 Hz." with the following text:

"A critical step in the setup for the numerical simulations consists in the choice of the frequency range. In order to reproduce accurately the wave propagation at high frequencies it is required a fine spatial and temporal sampling and therefore a larger computational effort. On the other hand, the simulation of wavelengths much shorter than the dimensions of the heterogeneities in the model would be out of scope."

Reviewer's remark (14)

P7/L27: The simulated signals are 65s long. Could it be that you are missing some late arrival (at time > than 1 minute after the OT)? Did you test this point?

Authors' answer: In order to support our choice of the simulated signal lengths we add the following sentence in the last paragraph of the subsection 3.2 "Setup for the computations": "We selected the length of the simulated seismograms in order to include all the significant signal for our purposes at the farthest station considered in the comparisons."

Reviewer's remark (15) P8/L26-28: I suggest to show in one figure an example, for one event and one or two stations, of the 3 compared signals (data, 3D, and 1D model) and the measure done on these signals (Arias duration , marking the 5% and 95% and PGV). I do not find very satisfactory the duration boxes showed in figure 10.

Authors' answer: As requested also by the other reviewer, we changed figure 10 with a plot showing a clearer comparison in terms of waveforms as well as Fourier amplitude spectra.

Reviewer's remark (16)

Figure 10: Why you do not show the Z component? And instead you show the Norm? "Empirical" are the Data, right? Of course we do not expect that the 3D model is able to match every wiggle of the data; however comparing data and synthetics from 3D model, it is clear that the arrival time and the number of energy packages in the data are not the same as the simulated ones. This might be mainly due, in my opinion, to wrong interfaces in the 3D model that causes or not, spurious arrivals. Also the first arrival appears to not fit the data. Can the author comment on that? Moreover the duration boxes plotted in the figures appear to have a strange starting and ending point, neglecting some shaking.

Authors' answer: In order to answer to the question about the Z component, which was posed also by the other reviewer, we add in paragraph 4.2, on page 8, the following text: "The vertical component was excluded from this comparison since it was systematically lower than the horizontal ones.". However we agree that the vertical components can be useful as complementary information in the waveform comparison and we include them in the improved Figure 10. Our attempt was not to match "every wiggle of the data" but to be able to predict the peak ground velocity and signal duration, as already explained in the answer to the Reviewer's remark #3. Concerning the spurious arrivals, the following comment was already present in the conclusions that (P.10 L.19-20): "Some persisting inconsistencies between the predicted and the observed data can be attributed to local errors in the 3D model as well as to errors in the assumed source parametrization for the simulated earthquakes." We do not think there is much more to be said about that. The duration boxes describe the time interval between 5% and 95% of the Arias intensity and therefore leave some shaking outside. In order to clearify this point to the reader we added adopted definition of the duration interval to the caption of the figure.

Reviewer's remark (17)

Figure 11-12-13: I would merge the 3 figures in a single panel, if possible.

Authors' answer: As requested, we merged the 3 figures in a single panel.

Reviewer's remark (18)

Figure 12, panel b): what is the wave package that appear to propagate from bottomleft to up-right, near the station T0824 (at its left), at the section top?

Authors' answer: The wave package evidenced by the Reviewer actually propagates from up-right to bottom-left and consists in the S-wave reflected from surface. In order to make this feature clear, we complement the panel with an additional S-wave "ray". In order to facilitate the reader to understand the snapshots of the simulated wavefield, we added as supplementary material the motion picture from which these snapshots were taken.

Reviewer's remark (19)

In the text sometimes the authors write "elongation of the ground motion". I would change it with long shaking duration or something like that.

Authors' answer: We have substituted the term "elongation of the ground motion" with "long duration of ground motion".

Reviewer's remark (20) It would be great, for the geophysical community, to have the model publicly available, in order to allow other scientists to compare models and to add, if possible new data.

Authors' answer: We agree with the Reviewer and therefore we make the model publicly available from Git-Hub and a pdf file with 3D content is provided as supplementary material.

Additional correction: table2: Computational cost on IBM-BG/Q = 50.000 core-hours